# A New Dual Fluorescence Method for Rapid Detection of Infectious Bronchitis Virus at Constant Temperature

**DOI:** 10.3390/microorganisms12071315

**Published:** 2024-06-27

**Authors:** Xinheng Zhang, Xiuhong Wu, Keyu Feng, Qian Wang, Qingmei Xie

**Affiliations:** 1State Key Laboratory of Swine and Poultry Breeding Industry, South China Agricultural University, Guangzhou 510642, China; xhzhang@scau.edu.cn (X.Z.); wuxiuhong@163.com (X.W.); fky19842004@163.com (K.F.); wwqq@stu.scau.edu.cn (Q.W.); 2Heyuan Branch, Guangdong Laboratory for Lingnan Modern Agriculture, Heyuan 517000, China; 3Guangdong Engineering Research Center for Vector Vaccine of Animal Virus, Guangzhou 510642, China; 4South China Collaborative Innovation Center for Poultry Disease Control and Product Safety, Guangzhou 510642, China; 5Key Laboratory of Animal Health Aquaculture and Environmental Control, Guangzhou 510642, China; 6Guangdong Provincial Key Lab of AgroAnimal Genomics and Molecular Breeding, College of Animal Science, South China Agricultural University, Guangzhou 510642, China

**Keywords:** infectious bronchitis virus, recombinase-aided amplification, detection at constant temperature

## Abstract

Infectious bronchitis virus (IBV) causes infectious bronchitis in chicken, an acute, highly contagious respiratory infection. Because of genetic mutations and recombination, IBV forms many subtypes, which makes it difficult to treat the disease and apply commercial vaccines. Therefore, to detect IBV in time and stop the virus from spreading, a novel and convenient IBV detection technology based on reverse transcription recombinase-aided amplification (RT-RAA) was established in this study. According to the *S1* gene of IBV CH I–V and Mass genotypes and *S1* gene of IBV CH VI genotype, a set of optimal primers were designed and selected to establish a real-time dual fluorescence RT-RAA method. The lowest detection line was 10 copies/μL of RNA molecules and the method exhibited no cross-reactivity with avian reticuloendotheliosis virus (REV), infectious bursal disease virus (IBDV), avian leukosis virus (ALV), Newcastle disease virus (NDV), chicken infectious anemia virus (CIAV), infectious laryngotracheitis virus (ILTV), Marek’s disease virus (MDV), and H9N2 avian influenza virus (H9N2), demonstrating high specificity. When compared to qPCR detection results, our method achieved a sensitivity of 96.67%, a specificity of 90%, and a Kappa value of 0.87 for the IBV CH I–V and Mass genotypes, and achieved a sensitivity of 100%, a specificity of 97.73%, and a Kappa value of 0.91 for the IBV CH VI genotype.

## 1. Introduction

Infectious bronchitis in chicken is a contagious respiratory disease. Clinical symptoms of this disease manifest as respiratory distress, nephritis, and reduced productivity. Initially, infectious bronchitis virus (IBV) causes symptoms such as coughing, sneezing and trachea rales. IBV mainly infects the ciliated epithelial cells of the respiratory tract and fallopian tubes, as well as the tubule cells of the kidneys, and can persist in hens for a long time, causing huge economic losses [1]. IBV belongs to the *Coronaviridae, Gammacoronavirus, Avian coronavirus*. IBV is a linear single plus-stranded RNA virus. When observed by electron microscopy, the virion has a radius of 45–100 nm and is spherical or oval in shape. IBV is an enveloped virus with spikes arranged in a crown shape on the surface of the virion.

From 1931, when IBV was first reported by Schalk et al., until now, more IBV genotypes have emerged due to gene recombination and mutation but with typical regional distribution [2]. With the economic globalization, under the background of China’s Belt and Road, international trade is increasingly prosperous, and the popularity and spread of IBV has also appeared across regions. From 1931 to 1985, different IBV strains in North America and Europe were mainly distinguished by clinical symptoms, including the Mass strain, Conne strain, Gray strain, Holte strain, Moroccan/G/83 strain, and other representative strains [3,4,5,6]. IBV has unique epidemic characteristics in Greece, Japan, South Korea, and other countries [7,8,9], as well as in China.

The Virus Neutralization Test (VNT) is considered to be the gold standard test for the detection of IBV serotypes. A single serotype of IBV inoculated per dose can produce highly specific antibodies that can distinguish genotypes and determine antibody titers in serum. However, there is a lack of standardization between the VNT system and operator, the VNT antigen and its preparation, and the mutation of the *S1* protein constantly produces new serotypes, making antigen preparation more difficult. Therefore, this method is suitable for laboratories but difficult to promote in clinical practice. Another serological method, ELISA, also has some clinical applications. An ELISA method targeting the IBV N protein achieved an accuracy rate of 93.06% [10]. However, because the detection time of antibody levels lags behind the early stage of the epidemic, there are certain limitations in clinical application. Molecular biological detection methods mainly include PCR, qPCR, and LAMP. Researchers have established PCR and real-time qPCR methods based on IBV’s *S1* gene for IBV monitoring [11,12]. Both PCR and qPCR have strong specificity but rely on professionals and variable temperature instruments, and PCR must be used in combination with gel electrophoresis equipment, so it is not suitable for clinical use in complex environments. Chandrasekar et al. established an RT-LAMP detection method based on IBV’s *S2* gene, and the detection could be completed at 60.0 °C for 45 min. This method was 100 times more sensitive than nested PCR [13]. LAMP does not require bulky instruments but it is prone to false positive, so the clinical use of this technology needs to be continuously optimized and improved.

Recombinase-aided amplification (RAA) is a breakthrough laboratory detection method that does not require expensive instruments. Using a portable fluorescence detector, this method can rapidly detect pathogens at 39~42 °C for 5–20 min. Researchers have established RAA methods for the detection of SARS-CoV-2, porcine Pseudorabies virus, H7 AIV, H5 AIV, duck circovirus, and other viruses [14,15,16,17,18,19]. RAA is also used in bacteria and parasites, such as clonorchis sinensis, schistosoma japonicum, E. coli, and Salmonella [20,21,22,23]. The advantages of RAA are that there is no need for fixed testing sites, the experimental instruments are easy to carry, the operation is simple, and the accurate results are immediately issued, which can save comprehensive costs and have broad prospects in the future.

This study establishes a dual fluorescence RAA detection method for IBV, providing a fundamental framework for simultaneously detecting various pathogen subtypes or employing multiple detection technologies. This method enables the rapid and precise on-site detection of IBV, opening up limitless possibilities for its application in point-of-care testing scenarios.

## 2. Materials and Methods

### 2.1. Virus

In our laboratory, we have preserved various avian viruses, including Newcastle disease virus (NDV, GenBank: JF950510), infectious bronchitis virus (IBV, GenBank: KR605489), infectious bursal disease virus (IBDV, GenBank: AF416621), infectious laryngotracheitis virus (ILTV, GenBank: JX458823), avian leukemia virus (ALV, GenBank: MT175600), subgroup H9N2 avian influenza virus (H9N2, GenBank: MN064851), and chicken infectious anemia virus (CIAV, GenBank: JX260426). The Marek’s disease virus (MDV, GenBank: L37202) was generously supplied by Harbin Pharmaceutical Group Bio-Vaccine Co., Ltd., based in Harbin, China. Additionally, avian reticuloendotheliosis virus (REV), not uploaded to GenBank, was provided by Wens Foodstuff Group Co., Ltd., located in Yunfu, China.

### 2.2. Reagents and Instruments

The pMD^TM^19-T Vector Cloning Kit (TAKARA, Guangzhou, China) was generously provided by Guangzhou Qiyun Bio & Tech Co., Ltd. (Guangzhou, China). For the RT-RAA nucleic acid Amplification Reagent (fluorescent type), we acquired it from Nanning Zhuangbo Bio & Tech Co., Ltd. (Nanning, China). The AxyPrep Body Fluid Viral DNA/RNA Miniprep Kit, essential for our work, was procured from Guangzhou Suyuan Biotechnology Co., Ltd. (Guangzhou, China). Throughout our experiments, we employed a real-time fluorescence quantitative PCR apparatus, specifically, the CFX96 model from Bio-Rad Laboratories, located in Shanghai, China. In this study, 50 throat swab samples suspected to be infected with IBV were provided by Wens Foodstuff Group Co., Ltd.

### 2.3. Design of Primers and Probe in RT-RAA

IBV’s *S1* gene sequences were downloaded from the NCBI database, and the GenBank number is shown in Table 1. The homology of IBV’s *S1* gene was analyzed by MEGAX64 (Mega Limited, Auckland, New Zealand) and DNASTAR (DNASTAR, Inc., Madison, WI, USA). IBV primers and probes were designed according to RAA primer and probe design principles. The primers and probe of IBV-VF/VR/VP were designed according to the *S1* genes of CH I-V and Mass genotypes, and the primers and probe of IBV-BF/BR/BP were designed according to the *S1* gene of CH VI *S1* genotype. Table 2 lists the specific sequences of the primers and probes. Additionally, Table 1 lists the reference strains used for IBV genotyping. The primers and probes used in this study were synthesized by Sangon Biotech Co., Ltd. (Shanghai, China).

### 2.4. Generation of Plasmid Standard

We designed the primers according to IBV’s *S1* gene, as shown in Table 3. The cDNA of IBV was used as a template for the PCR amplification. Takara’s pMD19-T kit was used to connect the target gene to the vector, and the product was transformed into DH5α receptor cells. The DH5α receptor cells were cultured and identified by bacterial solution PCR. PCR products with an expected band size were selected for sequencing. The bacterial solution with expected sequencing result was cultured, and we extracted the plasmid.

### 2.5. The RAA Method

RAA operates effectively at a temperature range between 39 °C and 42 °C, with amplification occurring within this constant temperature interval. The recombinase enzyme forms a complex with RAA primers, which then searches for particular binding sites on the target DNA. Simultaneously, a single-stranded binding protein is employed to open the double-stranded DNA, creating two single-stranded DNA strands. DNA polymerase subsequently initiates synthesis at the 3′ end of the primer, moving towards the 5′ end of the target DNA. This process generates new double-stranded DNA molecules, repeating the cycle for further amplification. When the exonuclease is inactive, the probe remains stable, and no fluorescent signal is emitted. However, when the probe binds to the target DNA, exonuclease activation occurs. Upon encountering the tetrahydrofuran (THF) moiety on the probe, exonuclease cleaves the link connecting the reporter group and the quenching group, releasing the reporter group to emit fluorescence. As the probe extends further, the blocker is removed, and the probe continues to elongate towards the 5′ end of the target DNA. This process enables the real-time collection and visualization of fluorescence signals. When the detection template is RNA, MLV reverse transcriptase can be added.

### 2.6. The RT-RAA Reaction System

The nucleic acids of 50 samples were extracted by the AxyPrep Body Fluid Viral DNA/RNA Miniprep Kit. After extracting, we preserved the nucleic acids at −80 °C.

The system of single fluorescence RT-RAA consisted of the following components: RT-RAA reaction unit (1 tube), PEG (25 μL), MgAc2 (2.5 μL), RAA-F (10 µM, 2 μL), RAA-R (10 µM, 2 μL), RAA-P (10 µM, 2 μL), DNA/RNA (2–5 μL), and ddH_2_O (11.5–14.5 μL) to make a total volume of 50 μL. On the other hand, the dual fluorescence RT-RAA system consists of: RT-RAA reaction unit (1 tube), PEG (25 μL), MgAc2 (2.5 μL), RAA-VF3 (10 µM, 1 μL), RAA-VR3 (10 µM, 1 μL), RAA-BF1 (10 µM, 2 μL), RAA-BR1 (10 µM, 2 μL), RAA-VP (10 µM, 0.4 μL), RAA-BP (10 µM, 0.6 μL), RNA (5 μL), and ddH_2_O (10.5 μL) to reach a total volume of 50 μL. The programmed conditions for both systems were as follows: 39 °C for 60 s initially, followed by 40 cycles of 41 °C for 30 s, with fluorescence signal collection. During this experiment, the evaluation of test sample results was based on Ct values, and the judgment criteria were as follows:

If the Ct value was ≤39 and an amplification curve occurred, the test sample was judged as positive. If the Ct value was >39 and no amplification curve occurred, the test sample was judged as negative.

### 2.7. Verification of RT-RAA Primers

The IBV-VF/VR specific primers were designed according to the regions with high conserved *S1* gene of CH I–V and Mass genotypes and were divided into 9 groups: IBV-VF1R1, IBV-VF1R2, IBV-VF1R3, IBV-VF2R1, IBV-VF2R2, IBV-VF2R3, IBV-VF3R1, IBV-VF3R2, IBV-VF3R2, IBV-VF3R3. Similarly, the IBV-BF/BR specific primers, designed based on the CH VI *S1* gene, were also divided into nine groups: IBV-BF1R1, IBV-BF1R2, IBV-BF1R3, IBV-BF2R1, IBV-BF2R2, IBV-BF2R3, IBV-BF3R1, IBV-BF3R2, and IBV-BF3R3.

The size of the products amplified by the primers were verified. Y10’s nucleic acid was used as positive control in the IBV-VF/VR primer group, CK/CH/GX/NN16-2 was used as positive control in the IBV-BF/BR primer group, and ddH_2_O was used as blank control. The RT-PCR system was composed of the following components: cDNA/DNA (2 μL), 2×Es Taq MasterMix (10 μL), PCR-F (10 µM, 1 μL), PCR-R (10 µM, 1 μL), and ddH_2_O (6 μL). Reaction condition: 95 °C for 3 min, followed by 35 cycles of 95 °C for 30 s, 58 °C for 30 s, and 72 °C for 10 s, concluding with 72 °C for 5 min. After PCR reaction, the products were analyzed by electrophoresis in 1% AGAR gel at 140 V for 20 min.

The 18 groups of fluorescent RAA primers were screened by a single fluorescence RT-RAA reaction. Y10’s nucleic acid was used as positive control in the IBV-VF/VR primer group, CK/CH/GX/NN16-2 was used as positive control in the IBV-BF/BR primer group, and ddH_2_O was used as blank control.

### 2.8. Specificity Test for the Single Fluorescence RT-RAA

To test the specificity of the fluorescence RT-RAA assay for IBV’s CH I–V and Mass genotypes, we used Y10, 4/91, LDT3-A, CKCHXJA09-1, TW2575/98, and H120 nucleic acids as positive controls, CH VI (CK/CH/GX/NN16-2), ALV, NDV, CIAV, ILTV, MDV, IBDV, REV, and H9N2 for the specificity analysis, and ddH_2_O as blank control for fluorescence RT-RAA nucleic acid amplification.

To test the specificity of the fluorescence RT-RAA assay for IBV’s CH VI genotype, we used CK/CH/GX/NN16-2 nucleic acid as positive control, Y10, 4/91, LDT3-A, CKCHJXJA09-1, TW2575/98, H120, ALV, NDV, CIAV, ILTV, MDV, IBDV, REV and H9N2 for the specificity analysis, and ddH_2_O as blank control for single fluorescence RT-RAA nucleic acid amplification.

### 2.9. Sensitivity Test for the Single Fluorescence RT-RAA

To establish the minimum detection limit of the fluorescence RT-RAA assay for IBV’s CH I–V and Mass genotypes, and IBV’s CH VI genotype, the plasmids constructed by the Y10 strain S gene and CK/CH/GX/NN16-2 strain S gene were diluted using a 10-fold ratio to 6 groups of plasmids with different concentrations: 10^0^, 10^1^, 10^2^, 10^3^, 10^4^, and 10^5^ copies/μL. A single fluorescence RT-RAA reaction was performed with 6 groups of different concentrations of plasmids as templates, and ddH_2_O was used as blank control. The test was repeated 3 times.

### 2.10. Specificity Test for the Dual Fluorescence RT-RAA

To verify the specificity of the dual fluorescence RT-RAA assay for IBV’s CH I–V and Mass genotypes and IBV’s CH VI genotype, nucleic acids of CK CH JX JA09-1, Y10, 4/91, LDT3-A, TW2575/98, H120, CK/CH/GX/NN16-2 were used as positive controls, and ALV, NDV, CIAV, ILTV, MDV, IBDV, REV, and H9N2 were used as blank controls for dual fluorescence RT-RAA nucleic acid amplification.

### 2.11. Sensitivity Test for the Dual Fluorescence RT-RAA

To verify the sensitivity, the plasmid constructed by the Y10 strain S gene and CK/CH/GX/NN16-2 strain S gene was mixed with a ratio of 1:1. This mixture was then diluted using a 10-fold ratio into 6 groups of plasmids with different concentrations: 10^0^, 10^1^, 10^2^, 10^3^, 10^4^, and 10^5^ copies/μL. A dual fluorescence RT-RAA reaction was performed with 6 groups of different concentrations of plasmids as templates, and ddH_2_O was used as blank control. The test was repeated 3 times.

### 2.12. Repeatability Test for the Dual Fluorescence RT-RAA

To investigate the sensitivity of the dual fluorescence RT-RAA assay for IBV’s CH I–V and Mass genotypes and IBV’s CH VI genotype at low template concentration, the plasmid constructed by the Y10 strain S gene and CK/CH/GX/NN16-2 strain S gene was mixed with a ratio of 1:1. This mixture was then diluted using a 10-fold ratio into 3 groups of plasmids with different concentrations: 10^1^, 10^2^, and 10^3^ copies/μL. A dual fluorescence RT-RAA reaction was performed with 3 groups of different concentrations of plasmids as templates, and ddH_2_O was used as blank control. The test was repeated 3 times.

### 2.13. Clinical Sample Testing

To verify that the dual fluorescence RT-RAA detection method established in this study could be used normally in the clinic, the dual fluorescence RT-RAA and qPCR methods were used to detect 50 clinical samples suspected of carrying IBV. Our method was evaluated by comparing and analyzing the results of qPCR and RT-RAA. The primer sequences of qPCR are shown in Table 3. The reaction condition of qPCR was as follows: 55 °C for 15 min, 95 °C for 30 s, 95 °C for 10 s, and 58 °C for 30 s, 40 cycles. The qPCR system consisted of the following components: RNA 2 μL, 2 × One Step U + Mix 10 μL, F (10 µM) 0.4 μL, R (10 µM) 0.4 μL, P (10 µM) 0.2 μL, and ddH_2_O 7 μL.

## 3. Results

### 3.1. Verification of RT-RAA Primers

The PCR verification results of the IBV CH I–V and Mass fluorescent RT-RAA detection primers are shown in Figure 1A. The PCR amplification products of the nine groups of IBV CH I–V and Mass fluorescent RT-RAA detection primers were all single specific bands without primer dimers and could be specifically bound to Y10’s nucleic acid. The size of the amplified product was consistent with the expected results. The screening results of the fluorescent RT-RAA detection method are shown in Figure 1B. All nine groups of IBV’s CH I–V and Mass genotypes of fluorescent RT-RAA detection primers had fluorescent signals when combined with the probe. Under the same concentration of detection templates, the primers of the IBV-VF3R3 group had the highest amplification efficiency. Based on the results of the PCR and fluorescent RT-RAA detection, the primers of the IBV-VF3R3 group were selected as the best primers for the detection of IBV’s CH I–V and Mass genotypes by fluorescent RT-RAA.

The PCR verification results of IBV’s CH VI fluorescent RT-RAA detection primers are shown in Figure 2A. The PCR amplification products of the nine groups of IBV’s CH VI fluorescent RT-RAA detection primers were all single specific bands without primer dimers and could be specifically bound to CK/CH/GX/NN16-2’s nucleic acids. The size of the amplified product was consistent with the expected results. The screening results of the fluorescent RT-RAA detection methods are shown in Figure 2B. All nine groups of IBV’s CH VI fluorescent RT-RAA detection primers had fluorescent signals when combined with the probe. Under the same concentration of detection templates, the primers of the IBV-VF1R1 group had the highest amplification efficiency. Based on the results of the PCR and fluorescence RT-RAA detection, the primers of the IBV-BF1R1 group were selected as the best detection primers for the fluorescence RT-RAA detection method of IBV’s CH VI genotype.

### 3.2. Specificity Test for the Single Fluorescence RT-RAA

The specificity results for IBV’s CH I–V and Mass genotypes are shown in Figure 3. Y10, 4/91, LDT3-A, CK CH JX JA09-1, TW2575/98, and H120 were the positive controls for IBV’s CH I–V and Mass genotypes, and all of them had fluorescence signals. Conversely, no fluorescence signal was observed for CH VI (CK/CH/GX/NN16-2) type, ALV, NDV, CIAV, ILTV, MDV, IBDV, REV, H9N2, and ddH_2_O. The results showed that primers and probe designed for IBV’s CH I–V and Mass genotypes could specifically detect nucleic acids of IBV’s CH I–V and Mass genotypes (representative strains Y10, 4/91, LDT3-A, CK-CH JX JA09-1, TW2575/98, and H120). There was no cross-reaction with nucleic acids of IBV’s CH VI (CK/CH/GX/NN16-2), ALV, NDV, CIAV, ILTV, MDV, IBDV, REV, and H9N2.

The specificity results for IBV’s CH VI genotype are shown in Figure 4. There were fluorescence detection signals in the CH VI (CK/CH/GX/NN16-2) positive control group. Y10, 4/91, LDT3-A, CK CH JX JA09-1, TW2575/98, H120, ALV, NDV, CIAV, ILTV, MDV, IBDV, REV, H9N2, and ddH_2_O showed no fluorescence signal. The results showed that the primers and probe designed for IBV’s CH VI genotypes in this study could specifically detect nucleic acids of IBV’s CH VI genotypes and could distinguish IBV’s CH I–V and Mass genotypes (Y10, 4/91, LDT3-A, CK CH JX JA09-1, TW2575/98, and H120), and had no cross-reaction with ALV, NDV, CIAV, ILTV, MDV, IBDV, REV, and H9N2.

### 3.3. Sensitivity Test for the Single Fluorescence RT-RAA Sensitivity

As shown in Figure 5, the lowest detection line of the fluorescent RT-RAA detection method for IBV’s CH I–V and Mass genotypes was 10^1^ copies/μL, and the detection rate was 100% in the 10^1^–10^5^ copies/μL group.

As shown in Figure 6, the lowest detection line of the fluorescent RT-RAA detection method for IBV’s CH VI genotype was 10^1^ copies/μL, and the detection rate was 100% in the 10^1^–10^5^ copies/μL group.

### 3.4. Specificity Test for the Dual Fluorescence RT-RAA

As shown in Figure 7, in the positive control group, Y10, 4/91, LDT3-A, CK CH JX JA09-1, TW2575/98, and H120 had FAM fluorescence (blue) signals, while CK/CH/GX/NN16-2 had HEX fluorescence (green) signal. There were no fluorescence signals for ALV, NDV, CIAV, ILTV, MDV, IBDV, REV, and H9N2. The results showed that our dual fluorescence RT-RAA method could specifically detect IBV’s CH I–V and Mass genotypes (FAM blue fluorescence signal), and specifically detect IBV’s CH VI genotype (HEX green fluorescence signal). There was no cross-reaction with ALV, NDV, CIAV, ILTV, MDV, IBDV, REV, and H9N2, showing strong specificity.

### 3.5. Sensitivity Test for the Dual Fluorescence RT-RAA

Copy numbers ranging from 10^0^ to 10^5^ copies/μL were utilized. The experiment was conducted three times using six groups with mixed concentrations, and ddH_2_O served as the blank control. As shown in Figure 8, the minimum detection limit for the IBV dual fluorescence RT-RAA detection method was 10^1^ copies/μL. Furthermore, the detection rate of 10^1^ to 10^5^ copies/μL was 100%. In other words, the IBV dual fluorescence RT-RAA detection method could detect RNA samples with concentrations exceeding 10 copies per microliter, demonstrating high sensitivity.

### 3.6. Repeatability Test for the Dual Fluorescence RT-RAA

As shown in Figure 9A, all three groups of low-concentration templates had fluorescence detection signals. The coefficient of Variation (CV) of Ct values of the three groups of IBV’s CH I–V and Mass genotypes with the fluorescence RT-RAA detection method were 4.20%, 2.81%, and 4.23%, respectively, all less than 10% (Figure 9B). The CV of Ct values of the three groups of IBV’s CH VI genotype with the fluorescence RT-RAA detection method were 4.86%, 4.25%, and 5.91%, respectively, all less than 10% (Figure 9C). The results showed that IBV’s dual fluorescence RT-RAA detection method still had good sensitivity at low concentration.

### 3.7. Results of the Clinical Sample Test

The results are summarized in Table 4. In this study, the Kappa value was used to judge the evaluation consistency. Compared with the qPCR method, IBV’s dual fluorescence RT-RAA method showed a sensitivity of 96.67%, a specificity of 90%, and a Kappa value of 0.87 (Kappa > 0.81, meaning almost perfect) for IBV’s CH I–V and Mass genotypes, and showed a sensitivity of 100%, a specificity of 97.73%, and a Kappa value of 0.91 for IBV CH VI genotype. The results showed that IBV’s dual fluorescence RT-RAA method was feasible for clinical application.

## 4. Discussion

Our team’s previous study showed that there were multiple IBV genotypes in the same area, so a vaccine for one genotype could not provide effective protection for chickens in the same area. For example, the vaccine of the H120 and 4/91 strains can provide greater than 65% protection against QX IBV but do not provide effective protection against TW IBV [24]. Therefore, it is not enough to prevent IBV by vaccine alone, but cutting off transmission routes is also a reliable preventive measure.

In China, IBV is more difficult to control due to different farming models and different regional development rates. Liu et al. monitored the prevalence of IBV in China for a long time and categorized the IBV strains into CH I–CH VII and Mass genotypes according to the *S1* gene sequence [25]. Our laboratory conducted an analysis of 209 strains of IBV isolated between 2004 and 2012, revealing that the QX type (also known as LX4 type or A2 type) accounted for 62% of these strains, making it the most prevalent genotype of IBV in China. This finding was consistent with the results of other researchers’ studies [26,27,28,29,30,31]. Between 2013 and 2015, we analyzed 290 strains of IBV collected from 13 provinces and cities in China. During that period, the QX type (CH I), Taiwan I (CH V), and 4/91 (CH II) accounted for 44.0%, 26.4%, and 13.0%, respectively. These results indicated that in addition to CH I, strains CH V and CH II were gradually becoming prevalent in China [32]. From 2016 to 2018, we analyzed 133 strains of IBV from the same 13 provinces and cities. During that period, 62.4% of the recombinant strains were CH I, and 9.8% were CH V. Furthermore, the analysis of recombinant strains revealed that their parents also belonged to CH I, CH V, and CH II [24]. In summary, the predominant IBV strains in China are CH I, CH V, and CH II, with CH V gradually increasing and becoming the dominant genotype in certain provinces.

In response to the complex and variable outbreaks of IBV, researchers have developed a number of detection methods. These methods include virus isolation, RT-PCR, RT-qPCR, RT-LAMP, ELISA, and IFA. However, these methods all have some shortcomings, such as RT-PCR and RT-qPCR relying on expensive and cumbersome instruments, and the primers of RT-LAMP being prone to false positives due to non-specific binding. It is difficult to prevent and control IBV because of the drawbacks of the above detection methods and the consumption of laboratory labor and resources. In order to improve this situation, this study designed fluorescent RT-RAA primers according to the IBV *S1* genotyping method [26], and established a dual fluorescence RT-RAA detection method.

The results of the RT-RAA specificity test showed that the primers of IBV-VF3R3 and the primers of IBV-BF1R1 had strong specificity, and there was no cross-reaction between the two groups of primers. IBV-VF3R3 primers could specifically detect IBV’s CH I–V and Mass genotypes (showing blue signal), and IBV-BF1R1 primers could specifically detect IBV’s CH VI genotypes (showing green signal).

In this study, the sensitivity of single fluorescence RT-RAA was consistent with that of dual fluorescence RT-RAA, both of which could detect 10^1^ copies/μL of RNA molecules. The sensitivity was better than that of the qPCR method established by Okino et al., which could detect 10^2^ copies/μL RNA molecules [33], and was consistent with the sensitivity of the LAMP method established by Liu Cheng [34]. In addition, the repeatability test of this study showed that the dual fluorescence RT-RAA detection method established in this study could maintain strong repeatability and accuracy under the condition of low detection template concentration.

The results of the clinical sample test showed that RT-RAA had the same sensitivity as qPCR and had good specificity and repeatability. Therefore, RT-RAA can be used in the clinic.

The IBV dual fluorescence RT-RAA method can be combined with a portable GENCHEK fluorescence detector. This would improve the reliability of inspection in complex and harsh environments. The RT-RAA method eliminates the need for highly specialized personnel, making it suitable for remote areas. Furthermore, it maintains the same sensitivity and accuracy as RT-qPCR. Importantly, this method can complete the detection within 30 min, and at the same time, the IBV strain can be classified to facilitate emergency vaccination to reduce the economic loss caused by an outbreak. In conclusion, the results indicate that fluorescent RT-RAA is a promising general detection technology, which provides reliable detection technology support for the clinical first-line detection of IBV in chickens.

## 5. Conclusions

The dual fluorescence RT-RAA assay for IBV genotyping established for the first time in this study can detect 10^1^ copies/μL of RNA molecules within 30 min. It offers numerous merits, including ease of operation, high sensitivity and specificity, excellent repeatability, and portability.

## Figures and Tables

**Figure 1 microorganisms-12-01315-f001:**
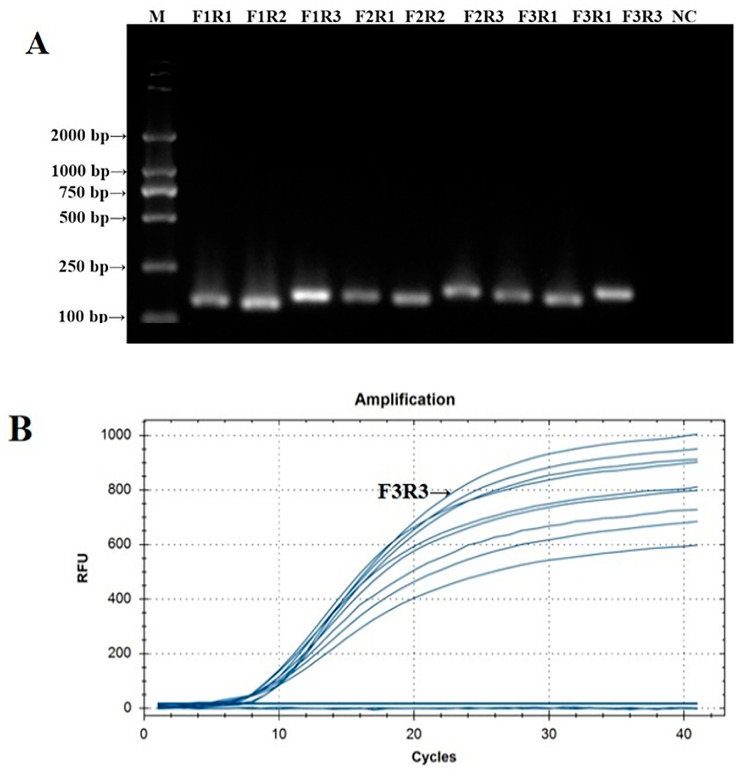
Validation and screening results of IBV’s CH I–V and Mass genotypes’ fluorescence RT-RAA detection primers. (**A**): PCR validation of the 9 groups of IBV’s CH I–V and Mass genotypes’ RT-RAA primers; (**B**): RT-RAA screening of the 9 groups of IBV’s CH I–V and Mass genotypes’ RT-RAA primers.

**Figure 2 microorganisms-12-01315-f002:**
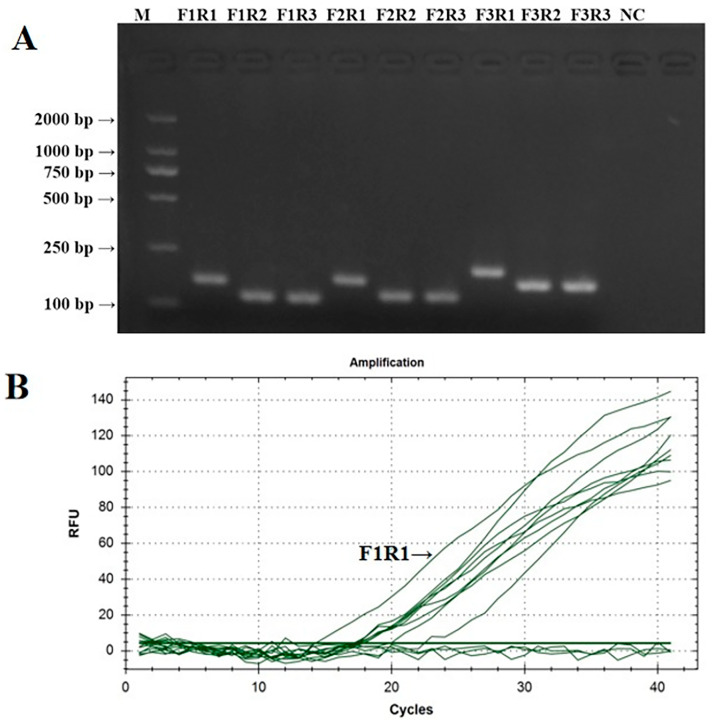
Validation and screening results of IBV’s CH VI genotype fluorescence RT-RAA detection primers. (**A**): PCR validation of the 9 groups of IBV’s CH VI genotype RT-RAA primers; (**B**): RT-RAA screening of the 9 groups of IBV’s CH VI genotype RT-RAA primers.

**Figure 3 microorganisms-12-01315-f003:**
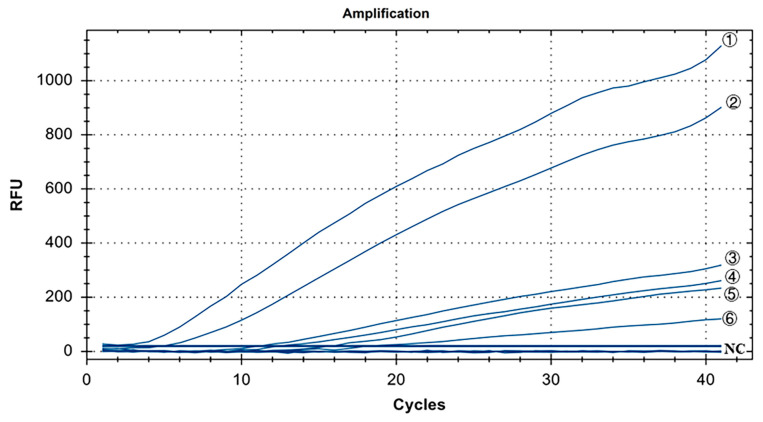
Specificity test results for single fluorescence RT-RAA of IBV’s CH I–V and Mass genotypes. ①–⑥: Y10, H120, LDT3-A, 4/91, CK CH JX JA09-1, TW2575/98; NC: CK/CH/GX/NN16-2, ALV, NDV, CIAV, MDV, IBDV, REV, AIV.

**Figure 4 microorganisms-12-01315-f004:**
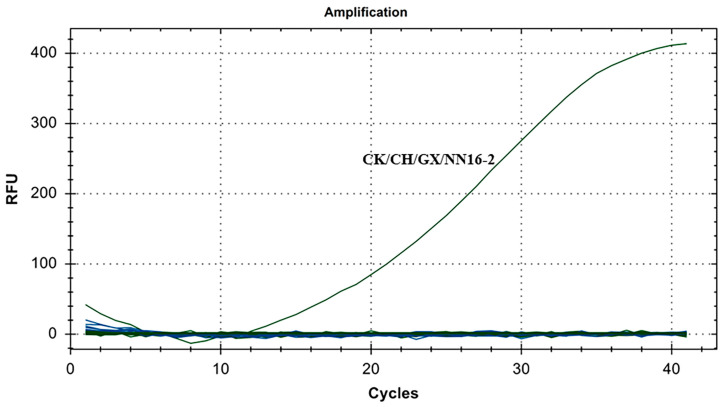
Specificity test results for single fluorescence RT-RAA of IBV’s CH VI genotype.

**Figure 5 microorganisms-12-01315-f005:**
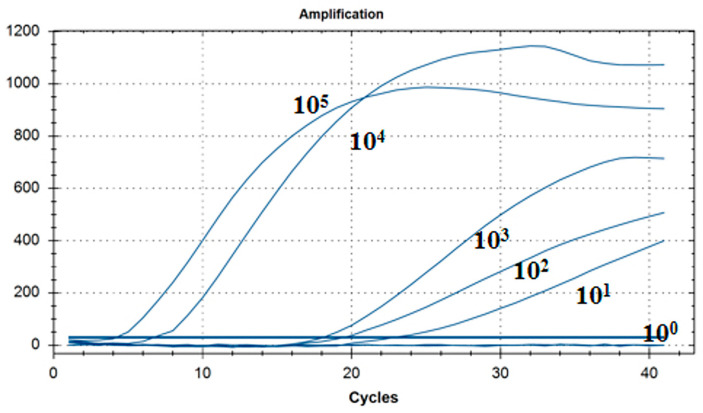
Sensitivity test results for single fluorescence RT-RAA of IBV’s CH I–V and Mass genotypes. The figure shows the optimal result of 3 repeated tests of different concentrations of plasmids.

**Figure 6 microorganisms-12-01315-f006:**
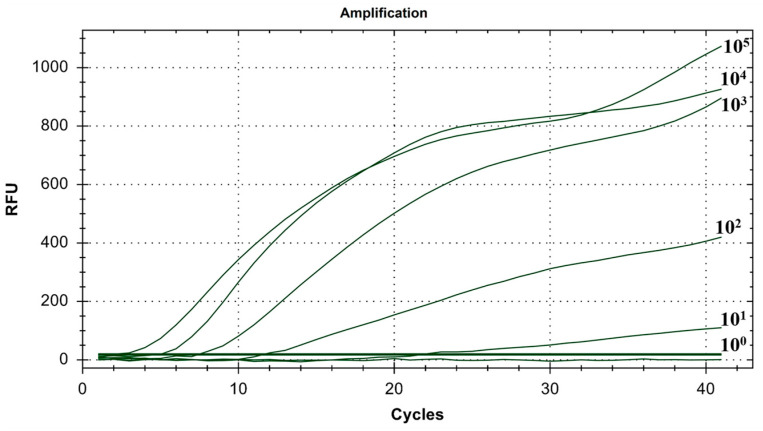
Sensitivity test results for single fluorescence RT-RAA of IBV’s CH VI genotype. The figure shows the optimal result of 3 repeated tests of different concentrations of plasmids.

**Figure 7 microorganisms-12-01315-f007:**
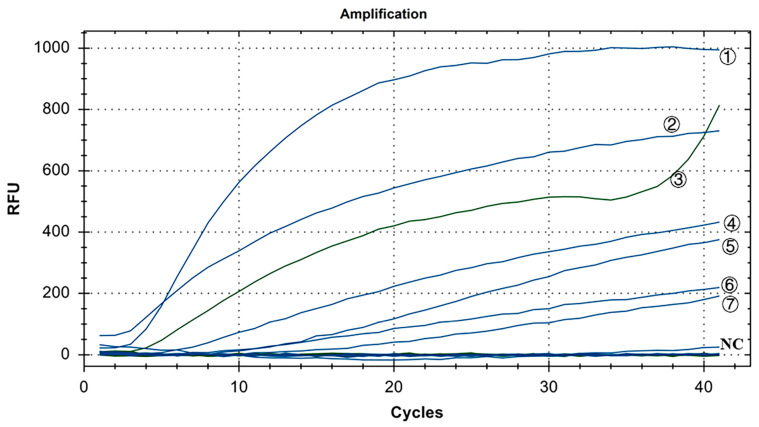
Specificity test results of IBV’s dual fluorescence RT-RAA assay. ①–⑦: Y10, H120, CK/CH/GX/NN16-2, LDT3-A, 4/91, CK CH JX JA09-1, TW2575/98; NC: ALV, NDV, CIAV, MDV, IBDV, REV, AIV.

**Figure 8 microorganisms-12-01315-f008:**
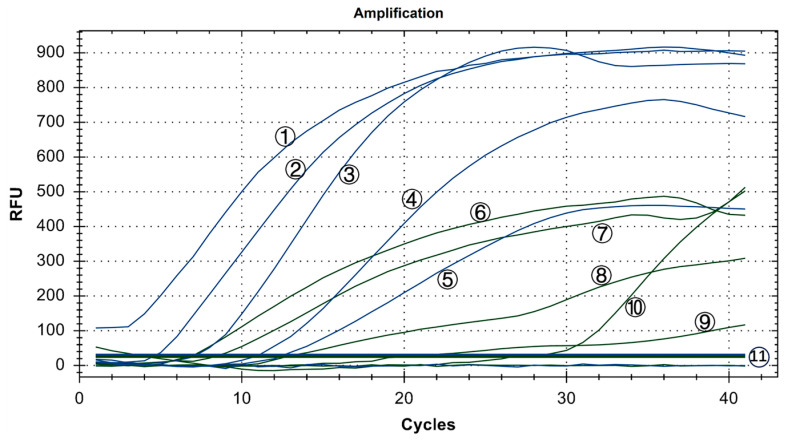
Sensitivity test results of IBV’s dual fluorescence RT-RAA assay. The figure shows the optimal result of 3 repeated tests of different concentrations of plasmids. ①–⑤: 10^5^–10^1^ copies/μL of IBV’s CH I–V and Mass genotypes’ standard plasmids; ⑥–⑩: 10^5^–10^1^ copies/μL of IBV’s CH VI genotype’s standard plasmids; ⑪: 10^0^ copies/μL of IBV’s CH I–V and Mass genotypes’ standard plasmids and CH VI genotype’s standard plasmids.

**Figure 9 microorganisms-12-01315-f009:**
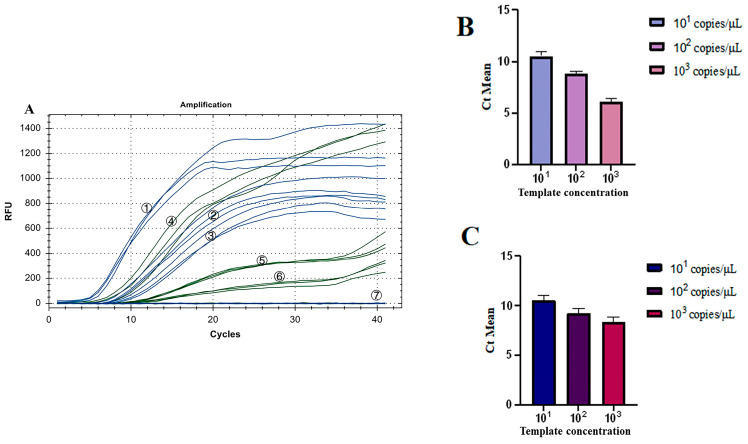
Repeatability test results of IBV’s dual fluorescence RT-RAA. (**A**): The results of 3 repeated tests of different concentrations of plasmids from 10^1^ to 10^3^copies/μL. ①–③: 10^3^–10^1^ copies/μL of IBV’s CH I–V and Mass genotypes’ standard plasmids; ④–⑥: 10^3^–10^1^ copies/μL of IBV’s CH VI genotype’s standard plasmids; ⑦: ddH_2_O. (**B**): Analysis of the repeatability of IBV’s CH I–V and Mass genotype’s dual fluorescence RT-RAA using Ct values for copy numbers ranging from 10^1^ to 10^3^ copies/μL, demonstrating a coefficient of variation (CV) of less than 10%. (**C**): Analysis of the repeatability of IBV’s CH VI genotype’s dual fluorescence RT-RAA using Ct values for copy numbers ranging from 10^1^ to 10^3^ copies/μL, demonstrating a coefficient of variation (CV) of less than 10%.

**Table 1 microorganisms-12-01315-t001:** The reference strains of IBV genotyping.

Name of Virus Strain	Genotype	Accession Number
CK/CH/GD/HY09	CH II	HQ018887.1
Y10	CH I	JX840411.1
CK/CH/GZHP/200108	CH I	MW560644.1
CKCHYNML1912	CH I	MT766988.1
IBV-SDTA-171217	CH I	MH159180.1
CK/CH/Guangdong/Heyuan3/0910	CH II	GU938421.1
LDT3-A	CH III	KR608272.1
CK/CH/GX/NN09 S1	CH III	HQ018900.1
CK CH SC ZJ12-1	CH III	KC692249.1
CK CH GX YL12-3	CH III	KC692265.1
4/91	CH II	KF377577.1
CK CH JX JA09-1	CH IV	HQ018890.1
GX-YL5	CH IV	FJ907238.1
CK CH GD LZ12-4	CH IV	KC692277.1
GD04-1	CH IV	DQ459473.1
TW2575/98	CH V	DQ646405.2
CK/CH/GD/GZ14	CH V	KX107660.1
GZ14F80	CH V	MG517474.1
CK/CH/YN/SL12-3	CH V	KJ524636.1
H120	Mass	KR605489.1
CK/CH/GX/NN16-2	CH VI	MF447729.1
H52	Mass	AF352315.1
THA320352	Mass	GQ885138.1
CK/CH/GD/XX1412-2	CH VI	KX107683.1
Ma5	Mass	AY561713.1
TC07-2	CH VI	GQ265948.1
CK/CH/GXNN09	CH VI	MT766940.1

**Table 2 microorganisms-12-01315-t002:** Sequences of primers and probes.

Primers	Sequences (5′-3′)
IBV-BF1	TTTGTTACACATTGTTTTAAAAATGGAC
IBV-BF2	TACACATTGTTTTAAAAATGGACAAGG
IBV-BF3	TTTAGTACAATTGTTGTTTTTGTTACACATTG
IBV-BR1	TAAATTTACTATAACTAGAAGTGGTAACTG
IBV-BR2	AATGTAATGATTTAAATTTACTATAAC
IBV-BR3	TTAACACAATGTAATGATTTAAATTTAC
IBV-BP	TAAATTAAGGGAGGGTGATATTCGTATTGG/iHEXdT//THF//iBHQ1dT/TCTAGATAGTAGTGG[C3-spacer]
IBV-VP	TAGGCCAAGGTTTTATTACAATGTGACTGAT/i6FAMdT//THF//iBHQ1dT/GCTGCTAATTTTAGT[C3-spacer]
IBV-VF1	TATTAATGCAACACAATTATAATAATATTAC
IBV-VF2	ATAAGTGTGTTGACTATAATATATATGGCAGAG
IBV-VF3	ATAAGTGTGTTGACTATAATATATATGG
IBV-VR1	AAGTATCTAAAATAGCTAACCCACCATCTGC
IBV-VR2	ATGGCACCCGAAGTATCTAAAATAGCTAACCCACC
IBV-VR3	AAGACCATAGCTGCCCTGTACAACAAAGACATC

**Table 3 microorganisms-12-01315-t003:** Primers’ sequences.

Primer Name	Sequence (5′-3′)	Amplified Fragment (bp)
PCR-IBVS-F	AAGACTGAACAAAAGACCGACT	1700
PCR-IBVS-R	CAAAACCTGCCATAACTAACATA	
qPCR-VF	GATGGCTCTCGTATACAGACTA	111
qPCR-VR	GCCTACTCTGCCATATATATTATAG	
qPCR-VP	AACGGAGCCCTTAGTATTAATGCAAC	
qPCR-BF	TGAGTGCGTATTGCTTGTGTTTATT	124
qPCR-BR	ATAATCAACACACCTATCTAAAACCACATT	
qPCR-BP	TAGCCCAGGCAGTCGCATATTTACTTCTGA	

**Table 4 microorganisms-12-01315-t004:** Test results of IBV’s dual fluorescence RT-RAA clinical samples.

		V-qPCR		B-qPCR	
		Positive	Negative	Positive	Negative
RT-RAA	Positive	29	2	6	1
	Negative	1	18	0	43
	Sensitivity (%)	96.67		100	
	Specificity (%)	90		97.73	
	Kappa	0.87		0.91	

## Data Availability

The original contributions presented in the study are included in the article material, further inquiries can be directed to the corresponding author.

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
