# Peer review of "A New Dual Fluorescence Method for Rapid Detection of Infectious Bronchitis Virus at Constant Temperature"

_microorganisms, 2024, doi:10.3390/microorganisms12071315_

Round 1
Reviewer 1 Report
Comments and Suggestions for Authors
In their article ”A new dual fluorescence method for rapid detection of infectious bronchitis virus by constant temperature“ Xinheng Zhang present a method for detection of IBV based on reverse transcription recombinase-aided amplification (RT-RAA). They used a set of primers designed according to the S1 gene of IBV CH Ⅰ-Ⅴ and Mass genotypes and S1 gene of IBV CH Ⅵ genotype, and they applied a real-time dual fluorescent RT-RAA method. Their results show a good sensitivity of 96.67% and a specificity of 90%.
In my opinion, the manuscript is very interesting, but I have a few notes.
Introduction.
The Introduction is too long. The paragraph between lines 57-73 is better to be in Section Discuses.
Materials and Methods.
2.5. Sample preparation. Тhe information from here can be added to the next one 2.6., because it is part of it. In 2.13. You describe more about samples, and I think that 2.5. is redundant.
Discussion.
Lines 365-367 you need to cite.
Lines 380-395 the information is not proper for the Section. You have to describe your results compared with other studies, not to explain methods. If you want to give more data for the RAA, you have to add it in Section Materials and Methods.
In conclusion, the section needs serious revision and further discussion.
Author Response
Thank you very much for your comments and suggestions. I have revised my manuscript according to your suggestions. All changes are highlighted in yellow.
Introduction.
I have transferred the paragraph between lines 57-73 to lines 370-386 in Section Discussion.
Materials and Methods.
2.5. Sample preparation. I have added this paragraph to 2.6 and create a new 2.5 to explain the RAA method.
Discussion.
Lines 365-367. Here I quoted Ms. Chen Tong's master's thesis, and I have corrected some mistakes in the previous quotation (line 492).
Lines 380-395. I have added this paragraph to lines 134-150 in 2.5.
If you have any questions, please feel free to contact me. Look forward to hearing from you.
12 June, 2024
Reviewer 2 Report
Comments and Suggestions for Authors
This paper presents a highly valuable study on the development of a novel and convenient detection technology for Infectious Bronchitis Virus (IBV). The study established a reverse transcription recombinase-aided amplification (RT-RAA) based detection technology. The authors have successfully designed and selected a set of optimal primers targeting the S1 gene of IBV CH I-V and Mass genotypes, and the S1 gene of IBV CH VI genotype. This real-time dual fluorescent RT-RAA method can detect as low as 10 copies/μL of RNA molecules, demonstrating high specificity with no cross-reactivity with other avian viruses. Overall, this reviewer recommends the acceptance of this paper.
Author Response
Thank you very much for your review and appreciation.
If you have any new suggestions, I'd be happy to be informed.
12 June, 2024
Reviewer 3 Report
Comments and Suggestions for Authors
Zhang et al. present a new detection method for infectuous bronchitis virus. It seems to work fine, asnd they also test clinical samples. Since the method allows reliable detection of IBV at low temperatures, it might be of interest to the readers. I just have some minor comments:
IBV only infects chicken, as the authors methion in the introduction. However, that fact is not mentioned in the abstract so a way higher relevance is suggested to the reader. Please mention in the abstract that IBV only infects chicken.
l. 57-73 explain CH I-V and Mass types before reporting their distributions. How are they differentiated?
How exactly and with which criteria were primers generated?
Omit figures 5B, 6B, 8B since they only show that detection works from 10^1 copies/ul which can just be stated in the text.
l. 358: explain kappa value before using it
l. 424 "101copies" -> "10^1 copies"
Author Response
Thank you very much for your comments and suggestions. I have revised my manuscript according to your suggestions. All changes are highlighted in yellow.
Abstract. I have added “in chicken” to line 19 in abstract.
- lines 57-73. The modification is in the line 372. My statement was wrong and I have corrected it. The classification of IBV genotypes is based on S1 gene.
The modification about primers is in lines 115-123. The RAA Primer design principles include a primer length of 30-35 nt and an amplification length of 100-200 bp (at least less than 500 bp). Other parts are similar to the principles of conventional primer design principles.
Figures 5B, 6B, 8B has been omitted.
- line 358. The explanation of Kappa has been added to lines 354-357.
- line 424. The error has been corrected.
If you have any questions, please feel free to contact me. Look forward to hearing from you.
12 June, 2024